# Effect of homophily and correlation of beliefs on COVID-19 and general infectious disease outbreaks

**Claus Kadelka**[1]*, **Audrey McCombs**[2]

**1** Department of Mathematics, Iowa State University, Ames, IA, United States of America, **2** Department of Statistics, Iowa State University, Ames, IA, United States of America

* ckadelka@iastate.edu

**Data Availability Statement:** The complete Python implementation of the 510 two models is available at Github at https://github.com/ckadelka/homophily-outbreaks.

## Abstract

Contact between people with similar opinions and characteristics occurs at a higher rate than among other people, a phenomenon known as homophily. The presence of clusters of unvaccinated people has been associated with increased incidence of infectious disease outbreaks despite high population-wide vaccination rates. The epidemiological consequences of homophily regarding other beliefs as well as correlations among beliefs or circumstances are poorly understood, however. Here, we use a simple compartmental disease model as well as a more complex COVID-19 model to study how homophily and correlation of beliefs and circumstances in a social interaction network affect the probability of disease outbreak and COVID-19-related mortality. We find that the current social context, characterized by the presence of homophily and correlations between who vaccinates, who engages in risk reduction, and individual risk status, corresponds to a situation with substantially worse disease burden than in the absence of heterogeneities. In the presence of an effective vaccine, the effects of homophily and correlation of beliefs and circumstances become stronger. Further, the optimal vaccination strategy depends on the degree of homophily regarding vaccination status as well as the relative level of risk mitigation high- and low-risk individuals practice. The developed methods are broadly applicable to any investigation in which node attributes in a graph might reasonably be expected to cluster or exhibit correlations.

## Introduction

Infectious disease outbreaks have been on the rise for several decades, and account for more than one in eight deaths globally [1]. A comprehensive study of an infectious disease outbreak such as the current COVID-19 pandemic must involve not only the biological properties of the disease and its causal pathogen but also the societal circumstances affecting disease spread. Classical differential equation models assume homogeneous mixing (i.e., random contacts) of individuals and fail to account for the occurrence of increased interactions among people with similar beliefs or circumstances, a clustering phenomenon known as homophily. Network

**Funding:** The author(s) received no specific funding for this work.

**Competing interests:** The authors have declared that no competing interests exist.

models, on the other hand, allow for the study of the effect of homophily on disease outbreaks by explicitly incorporating clustering of individuals with similar beliefs or circumstances into social or physical interaction networks. Homophily among individuals who choose not to vaccinate for a variety of reasons (religious beliefs, fear of side effects, etc.) has been widely observed [2–4], and this type of homophily has been associated with both more frequent and larger disease outbreaks than would be expected under homogeneous or random social mixing [5–7]. Models that assume homogeneous mixing are therefore likely too optimistic and underestimate the level of vaccination required to achieve herd immunity and avoid outbreaks [8].

While the epidemiological implications of homophily regarding vaccination status have been well-studied, the effect of homophily with respect to other beliefs or circumstances, such as trust in the effectiveness of social distancing measures or risk status, has not received as much attention [9]. Especially during a pandemic, when risk mitigation measures are implemented globally, homophily regarding various beliefs or circumstances and population-wide correlations between them may have profound effects on disease spread. The correlation between social beliefs and partisan identification has been increasing in the United States [10], and ideological overlap between the two major political parties has diminished [11]. Recent polls by Gallup indicate that, in the United States, Democrats compared to Republicans are more likely to engage in risk mitigation (wearing masks [12], avoid eating out [13], avoid flying [14], etc.) and receive a COVID-19 vaccine [15]. Given the increases in both opinion polarization and correlations among opinions, as well as the effect homophily can have on disease dynamics, further work on the epidemiological consequences of correlated and clustered beliefs or circumstances is warranted.

Generating a network with *a priori* specified homophily and correlation among node attributes (belief in the safety of a COVID-19 vaccine, belief in the effectiveness of social distancing, etc.) can be technically challenging. In this study, we present a novel technique for applying binary attributes with a pre-defined correlation structure to a physical interaction network that exhibits a pre-defined level of homophily for each attribute. Using this technique, we investigate how homophily and correlations among several beliefs and circumstances affect the spread of an infectious disease in a Watts-Strogatz small-world physical interaction network (a community, a city, etc.) [16], and substantially influence the outcome of epidemiological studies and their predictions (Fig 1). First, we consider a simple, agent-based compartmental infectious disease model in which each agent has two binary belief attributes: confidence in vaccines and attitude toward social distancing measures. That is, a person either agrees to be vaccinated or not, and a person either engages in enhanced risk reduction (social distancing, mask wearing, etc.) or not. Then, in a more complex model developed specifically for COVID-19 (which includes hospitalization, asymptomatic carriers and differential risk status) [17], we add a third binary attribute distinguishing between high- and low-risk individuals. In the simple model, we focus on the frequency of an infectious disease outbreak (defined as >1% becoming infected) as primary outcome measure, while in the COVID-19 model, we focus on the number of deaths.

## Results

The generic infectious disease model yielded several expected results. An increase in the proportion of vaccinated, an increase in the proportion of distancers, an increase in the vaccine effectiveness, and an increase in the level of distancing practiced all led to lower disease burden, quantified by the disease outbreak frequency (S1 Fig) as well as the initial basic reproductive number $R_0$ (S2 Fig). The two outcome measures were highly positively correlated;

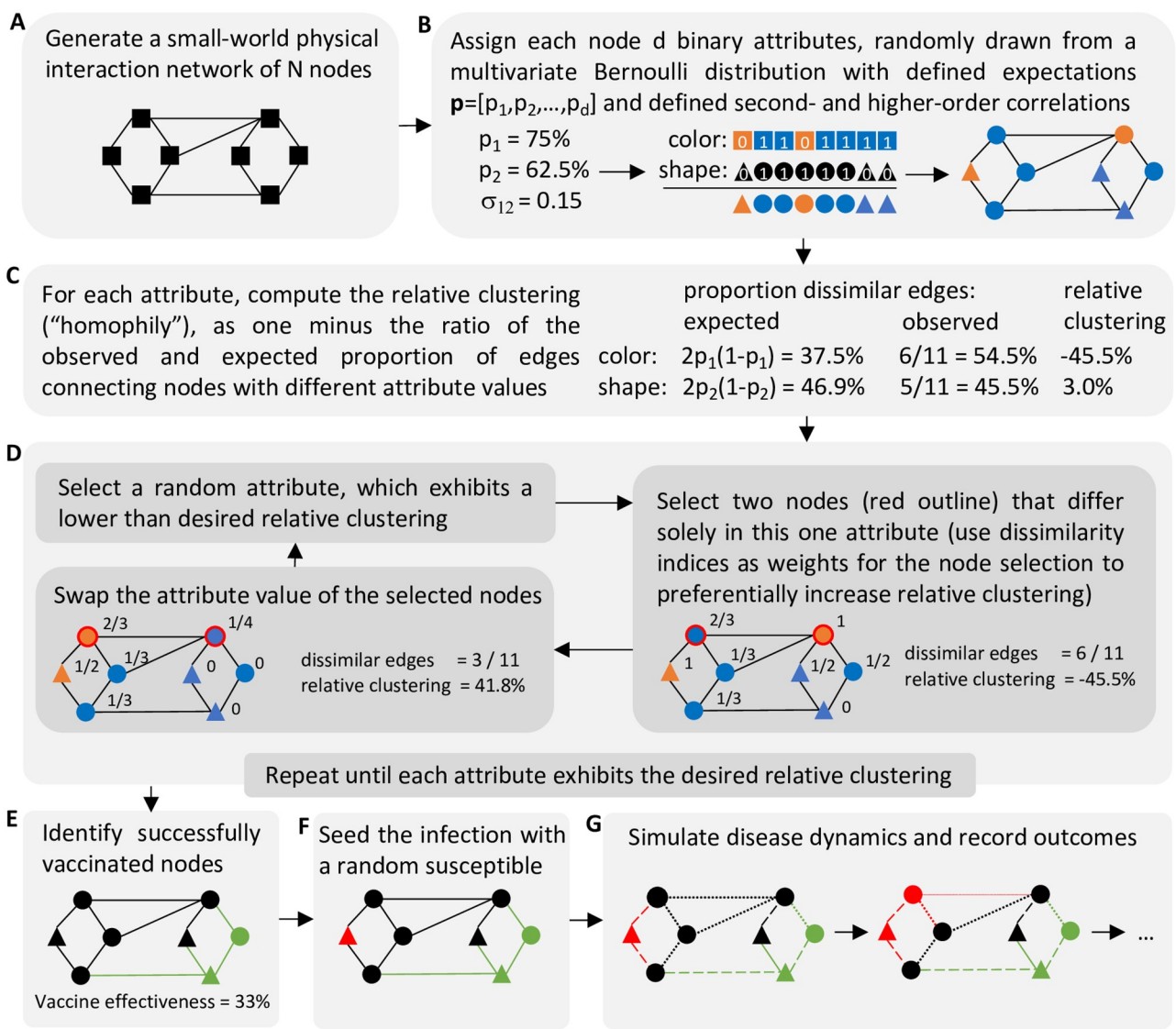

**Fig 1. Graphical overview of a simulation run.** (A) Generation of a physical interaction network, (B) Assignment of *d* correlated binary attribute values to each node. If *d* = 2, the attributes represent e.g. attitude towards vaccines (color) and social distancing (shape), (C) Computation of the relative clustering level (a measure of homophily) of each attribute, (D) Overview of the clustering algorithm used to assign attributes so that the network exhibits a desired level of homophily for each attribute. Values to the right of each node indicate its dissimilarity index: the proportion of neighbors with a different attribute value. (E) Vaccination of all nodes with a positive (blue) attitude towards vaccines and removal of those successfully vaccinated (green) from the pool of susceptible individuals; the probability that an all-or-nothing vaccine awards protection equals its effectiveness, (F) Infection of a randomly selected susceptible node (red), (G) Simulation of the spread of the infection and recording of outcomes. The likelihood of interaction (edge weight) depends on whether nodes practice social distancing (circles) or not (triangles).

therefore, we focused primarily on the outbreak frequency, which in network models incorporates more information than $R_0$.

Outbreaks occurred more frequently in interaction networks exhibiting homophily regarding vaccination and social distancing than those without homophily (Fig 2 and S1 Fig). In order to obtain a detailed understanding of the effect of homophily and correlation of opinions on the outbreak frequency, we fixed the values of several model parameters: we considered a situation where 2/3 of individuals vaccinate and 2/3 reduce their social contacts by 50%. As expected, outbreaks occurred less frequently in the presence of a more effective vaccine

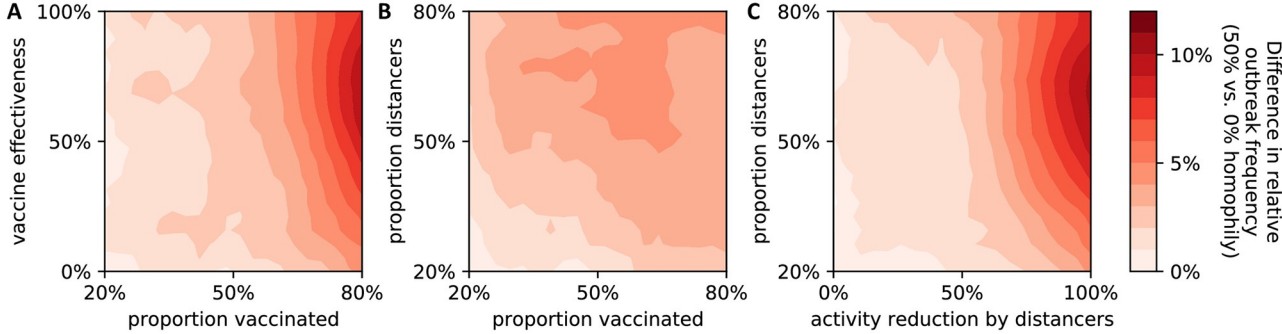

**Fig 2. Comparison of outbreak frequency in networks with and without homophily.** Contour plots were generated from 10,000,000 independent simulation runs with four vaccine and social distancing parameters chosen uniformly at random (axes show parameter ranges). The difference in outbreak frequency (where an outbreak was defined as >1% of the population eventually becoming infected) from a reference scenario of no vaccine and no social distancing was calculated for two scenarios: social interaction networks with 50% homophily of those who vaccinate and of those who practice distancing and networks without homophily (see S1 Fig). Data was binned and smoothed using a two-dimensional Savitzky-Golay filter [18] (details in Methods). Each subplot shows the effect of variation of two parameters on the difference in outbreak frequency between the two different homphily scenarios (see S1 Fig). (A) vaccine coverage (x-axis) and vaccine effectiveness (y-axis), (B) vaccine coverage (x-axis) and proportion of those who distance, (C) contact reduction (in %) by those who practice social distancing (x-axis) and proportion of those who distance (y-axis). An equivalent analysis for the basic reproductive number is shown in S2 Fig.

(Fig 3A). Contact networks in which those who vaccinate were also more likely to socially distance (correlation = 0.45) exhibited more outbreaks, while a negative correlation (−0.45) between vaccination and distancing led to fewer outbreaks than a situation with no correlation (Fig 3B). Contact networks in which those who vaccinate cluster were relatively more likely to experience an outbreak, especially in the presence of a highly effective vaccine (Fig 3C). Similarly, homophily in those who distance led to more outbreaks, but the effect was smaller (Fig 3D). Moreover, this effect became stronger as the correlation between vaccinated and distancers increased, and was absent when the correlation was negative. Interestingly, homophily in distancers led to slightly more outbreaks than no homophily in situations "without" a vaccine (effectiveness = 0%). In this case, as expected, it did not matter if those who received the vaccine clustered or not.

After being vaccinated individuals may choose to increase their level of social contacts because they believe they are immune, a phenomenon known as risk compensation [19]. We considered a model scenario in which those who vaccinate increase their activity levels on average by up to 41%. The release of an ineffective vaccine coupled with increased activity levels led to more outbreaks than a situation without a vaccine (region to the left of the black line in Fig 4). Presence (Fig 4A) or absence (Fig 4B) of clustering and correlation of those who vaccinate and those who distance did not affect the level of effectiveness needed so that a vaccine is beneficial. Neither, did these findings depend on the particular choice of the proportion of those who vaccinate and those who distance (S1 Table).

An important characteristic of COVID-19 is the increased risk of severe disease and death for older adults and people with comorbidities [20]. A recent COVID-19 model captures this differential risk by distinguishing between low-risk (2/3 of all individuals in the United States) and high-risk individuals [17]. Here, we adapted this model to investigate the effects of both increased contact between individuals of the same risk group, as well as differential vaccination coverage and social distancing levels between risk groups. As the basic reproductive number and the outbreak frequency both focus on disease transmission and fail to describe the differential risk of severe disease and death, for this analysis we focused instead on the number of deaths due to COVID-19.

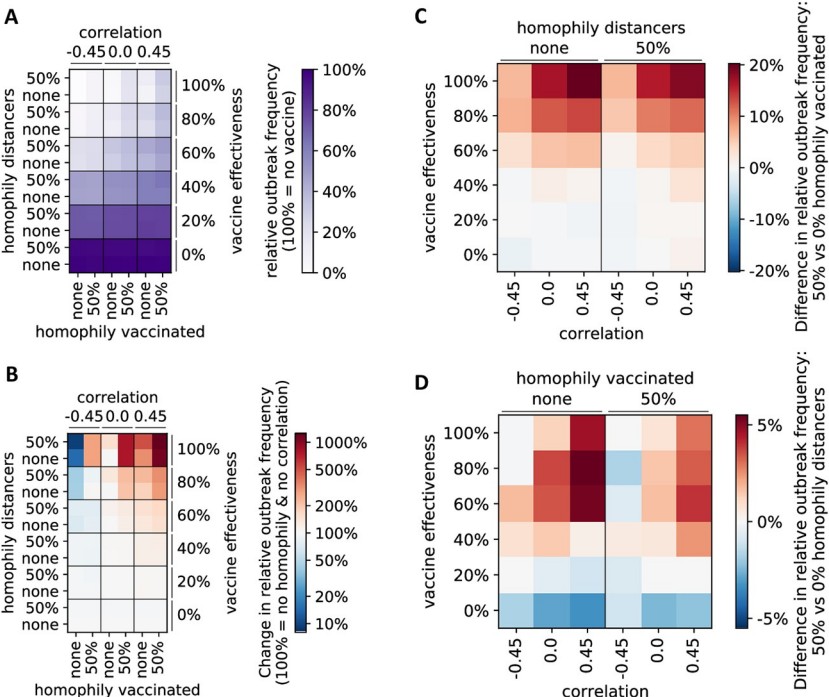

**Fig 3. Effect of homophily and correlation of opinions on outbreak frequency.** (A) The relative outbreak frequency is compared for different scenarios with respect to homophily and correlation of those who vaccinate and those who distance, and for different levels of vaccine effectiveness. Reference level for comparisons is a vaccine with 0% effectiveness and no homophily nor correlation of vaccinated and distancers. This reference level is set to 100%. (B) For each level of vaccine effectiveness, the change in relative outbreak frequency is compared to the homogeneous case of no homophily and no correlation, which is set to 100%, respectively. (C-D) Absolute difference in relative outbreak frequency (from A) when comparing physical interaction networks where (C) vaccinated, (D) distancers cluster (homophily = 50%) versus networks without homophily.

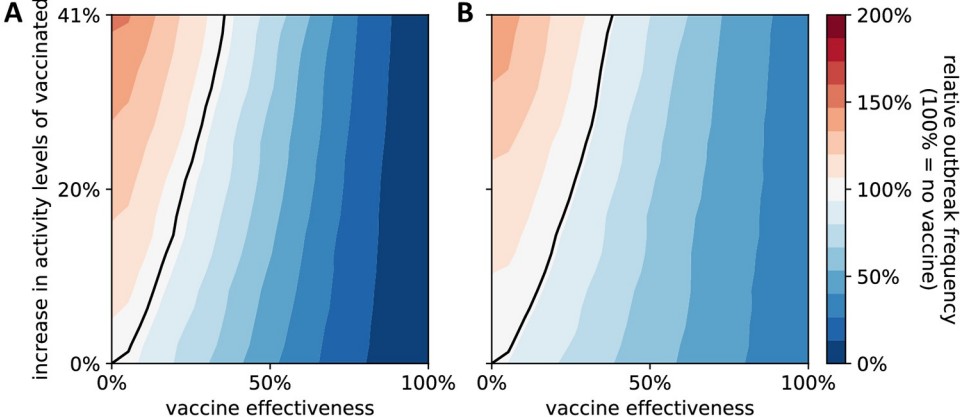

**Fig 4. Effect of increased activity levels by individuals who have received a vaccine.** The outbreak frequency (relative to the reference case of no vaccine, which is set to 100%) is shown for different levels of vaccine effectiveness (x-axis) and increased average activity levels by those who received a vaccine (y-axis). A black line depicts the x,y-coordinates at which the presence of the vaccine does not change the outbreak frequency. To the left (right) of this line, the presence of the vaccine is detrimental (beneficial). Two different scenarios regarding homophily and correlation of those who vaccinate and those who distance are considered: (A) 0% homophily and no correlation, (B) 50% homophily of those who vaccinate and those who distance and 0.45 correlation. In both plots, a fixed proportion of 65% of all individuals receive a vaccine and 65% of all individuals practice social distancing, i.e., reduce their social contacts by 50%. Data was binned and smoothed using a two-dimensional Savitzky-Golay filter [18] (details in Methods). See S1 Table for a sensitivity analysis where these proportions are varied.

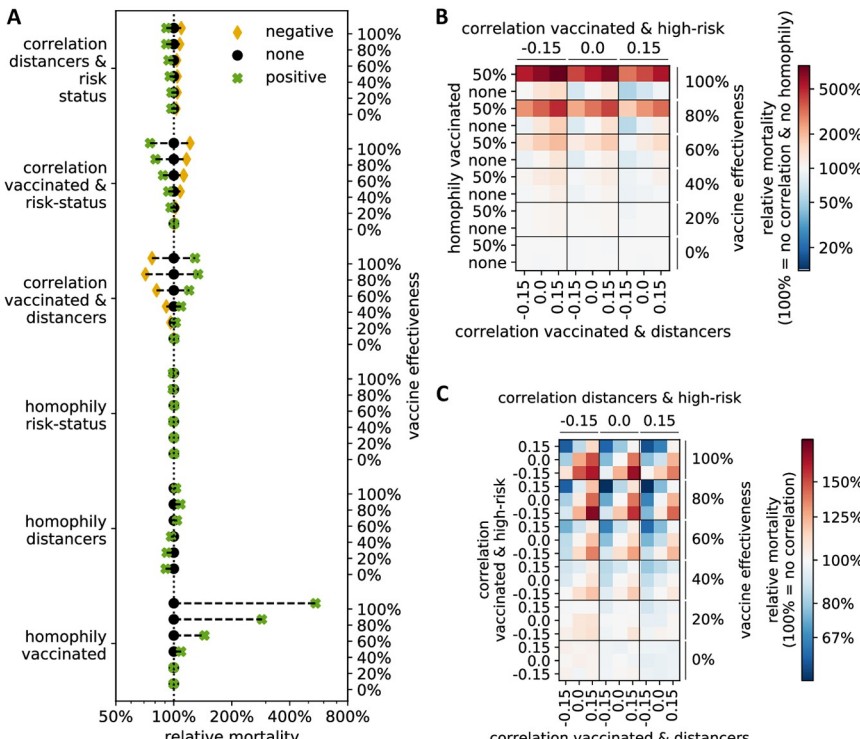

**Fig 5. Relative mortality in the COVID-19 model compared to the homogeneous case of no homophily and no correlation.** (A) For each line, the vaccine effectiveness and one homophily or correlation variable is fixed at a negative (−0.15 correlation; yellow diamond), positive (0.15 correlation or 50% homophily; green cross) or zero value (black circle), and average mortality is calculated across all other values and compared to the homogeneous case of no homophily and no correlation (dotted line; relative mortality = 100%). (B-C) Relative mortality is shown when three variables and the vaccine effectiveness are fixed. Red (blue) indicates higher (lower) mortality than in the homogeneous case of no homophily and no correlation. In (B) the three most influential variables from (A) are fixed, while in (C) the three correlations are fixed.

In the case of a highly effective vaccine, clustering of those who vaccinate remained the most important variable: with a perfect vaccine (100% effective), almost 600% more people die when comparing 50% with 0% homophily of vaccinated individuals (Fig 5A and 5B). Clustering of distancers had only a small effect on mortality, and the direction of the effect was dependent on the vaccine effectiveness: In the presence of a bad vaccine (effectiveness ≤40%), slightly fewer people died when those who practiced social distancing clustered, while the opposite was true in situations with a more effective vaccine (effectiveness ≥60%). If those who vaccinated were also more likely to practice social distancing (correlation = 0.15), more deaths occurred, while mortality was reduced if individuals who did not get vaccinated were more likely to practice social distancing (correlation = −0.15). Whether or not high-risk individuals clustered had very little effect on the number of deaths. However, the correlations between risk status, vaccination and social distancing proved important (Fig 5A–5C). With increasing vaccine effectiveness, overall mortality decreased when high-risk individuals were more likely to get vaccinated and practice social distancing, although the latter effect was weaker.

Empirical studies in multiple countries indicate that older people have fewer daily physical interactions than younger people [21, 22]. As older people make up a large proportion of the group of high-risk individuals it makes sense to assume a lower contact rate, or alternatively a higher contact reduction rate for high-risk individuals compared to low-risk individuals. In

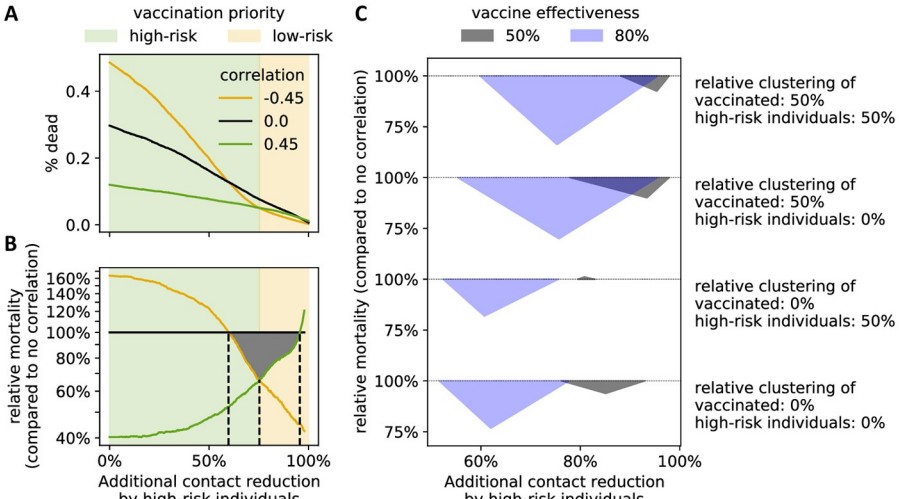

**Fig 6. Level of contact reduction by high-risk individuals influences vaccination priorities.** (A) The average mortality at a given additional contact reduction by high-risk individuals is shown for three different scenarios: negative (−0.45; yellow), zero (black) and positive (0.45; green) correlation between vaccinated and high-risk individuals. (B) Relative mortality compared to the case of no correlation (black line in A), at 50% homophily of both high-risk individuals and individuals who vaccinate. Black dashed lines and a gray triangle highlight the three intersection points of the three curves. (A-B) homophily of those who vaccinate and of high-risk individuals: 50%, vaccine effectiveness: 80%. Background colors indicate the prioritization (high-risk or low-risk individuals) that leads to lower overall mortality. (C) The location of the intersection points from (B) is shown for all combinations of homophily of those who vaccinate (0% vs 50%) and of high-risk individuals (0% and 50%), as well as two levels of vaccine effectiveness: 50% (gray) and 80% (blue). S4 Fig contains the full curves for all eight considered combinations.

addition to this inherent demographic difference, during the current pandemic contact reduction may be enhanced by additional social-distancing choices related to the higher perceived risk a COVID-19 infection presents for this vulnerable population. Without high-risk contact reduction, deaths can be averted by prioritizing the vaccination of high-risk individuals (Fig 5). To study the level of additional high-risk contact reduction at which the optimal vaccination strategy changes, we compared COVID-19-related mortality under three scenarios: high-risk individuals are more, equally, and less likely to vaccinate than low-risk individuals. As expected, a reduction of contacts by high-risk individuals led to fewer deaths under all scenarios (Fig 6A). The rate at which deaths decreased varied significantly, however: if high-risk individuals strongly reduced their contacts, vaccination of proportionately more low-risk individuals led to fewer deaths (Fig 6B).

We expected to find a single value for the additional contact reduction of high-risk individuals at which the correlation between who vaccinated and risk group did not matter. Instead, we found a range of values for which both negative and positive correlations yielded fewer deaths than no correlation. That is, a homogeneous vaccination strategy across risk groups (correlation = 0) was never optimal. For instance, in a situation with 80% vaccine effectiveness and 50% homophily of those who vaccinate and of high-risk individuals, we found that if high-risk individuals have 75% fewer interactions than low-risk individuals, both increased vaccination of low- or of high-risk individuals (correlation = 0.45) led to 34% fewer deaths than a homogeneous vaccination strategy (Fig 6B). As with other model results, the level of homophily of vaccinated individuals mattered, as the switch-point for optimal vaccination strategy was lower at 0% homophily of vaccinated versus 50% homophily. Furthermore, the switch-point was higher at lower vaccine effectiveness (Fig 6C). Interestingly, in the case of 0% homophily of vaccinated, 50% homophily of high-risk individuals, and a weak vaccine

(effectiveness = 50%), if high-risk individuals reduced their contacts by 81% more than low-risk individuals, then the distribution of the vaccine across the risk groups did not affect mortality. This was the only parameter setting of those we considered that matched our original expectation.

## Discussion

Contact between similar people occurs at a higher rate than among dissimilar people, a clustering phenomenon known as homophily [23]. While there is plenty of evidence that people with a negative view of vaccines cluster [4, 24, 25], we still lack empirical data on the degree of homophily regarding social distancing in response to the COVID-19 pandemic. There are, however, studies examining the average number of daily contacts stratified by age that show increased activity levels between people of similar age, which, given the strong correlation between COVID-19 risk status and age, suggests the presence of homophily regarding risk status. The correlation between policy views and partisan identification has been increasing [10], with more people now on the left or the right and fewer holding a mix of positions [11]. Recent polls by the Gallup agency indicate that an individual's willingness to receive a COVID-19 vaccine [15] and to practice social distancing (wear a mask [12], eat out [13], fly [14], etc.) differs substantially by age as well as political party affiliation. Older people (age 65+), who are more likely at high risk, practice more social distancing and are slightly more willing to get vaccinated against COVID-19. Democrats are much more likely to receive the vaccine and to practice social distancing, which suggests a positive correlation between these two attributes considered in our study. We do not have accurate estimates for the degree of homophily or the correlation among different attitudes related to the spread of an infectious disease generally, or COVID-19 specifically. We therefore studied the spread of a disease across a social network under different possible scenarios for homophily and correlation of beliefs or circumstances.

We found that the presence of homophily in opinions regarding whether to vaccinate or not, and whether to practice social distancing or not, as well as a large overlap between those who distance and those who vaccinate, can dramatically increase the probability of a disease outbreak, especially in the presence of a highly effective vaccine. Accordingly, any results obtained using classical differential equation models, which inherently assume homogeneous mixing and account for neither homophily nor correlation in opinion patterns, likely present lower bounds on the expected severity of an outbreak. Furthermore, if opinions are positively correlated and if real interaction networks exhibit even a modest degree of homophily, as the Gallup surveys suggest they do, our current social context corresponds to a scenario with a substantially worse disease burden than a homogeneous scenario (Figs 3 and 5, S3 Fig).

Our study produced several results relevant to policy makers. First, whether vaccination of low- or high-risk individuals should be prioritized depends on the relative contact rate of low-versus high-risk individuals. If high-risk individuals have substantially fewer contacts (Fig 6C), prioritizing the vaccination of low-risk individuals reduces overall mortality compared to homogeneous vaccination or vaccination of high-risk individuals. The reason for this is likely that increased vaccination of low-risk individuals, who are more socially active, can prevent outbreaks and reduce mortality of the vulnerable, less-vaccinated population. An empirical study in eight European countries revealed that prior to the pandemic older adults (65+) already had on average 41.8% fewer contacts than those age 65 and under [21] (S2 Table). Together with the increased perceived risk COVID-19 presents to the high-risk group, these individuals might indeed practice enough social distancing so that prioritizing the vaccination of low-risk individuals is optimal. Policy makers therefore might consider a heterogeneous approach, in which vaccination of high-risk individuals is prioritized in communities where

social distancing among the high-risk population is less prevalent. Our findings highlight the importance of accurate estimates of the contact and risk reduction practiced by individuals of different risk groups. Second, the optimal vaccination strategy in our model was affected by the homophily among those who vaccinate. For a given vaccine effectiveness, the level of additional contact reduction at which the optimal vaccine strategy switches from prioritizing high-risk to prioritizing low-risk individuals is higher in the presence of homophily (Fig 6C).

Third, if vaccinated individuals engage in risk compensation, i.e., increase their activity levels, the level of effectiveness a vaccine needs to possess so that it does not lead to a worse disease burden increases in the presence of homophily and correlation, but only marginally. Finally, we note that almost all effects (both positive and negative) described in this study became stronger as the effectiveness of the vaccine increased. This may be partially due to the fact that situations with a vaccine effectiveness of 100% are on average closest to the herd immunity threshold of $R_0 = 1$ (S2 and S3 Figs), as previous studies suggest the effect of homphily on outbreak probability is strongest when vaccination coverage is close to this threshold [5, 26]. Nevertheless, our results are counter-intuitive in this respect: While a more effective vaccine is certainly better in general, relative differences in outcomes and any negative effects due to the presence of homophily and/or correlation will be larger with a more effective vaccine.

We considered binary attributes and only second-order interactions among the attributes for two reasons: First, statistical methods for generating correlated Bernoulli random variables are well-established; and second, higher-order correlations lack an intuitive interpretation. Extending the model to include more binary attributes or higher-order interactions is straightforward using the algorithm we developed here. More complex scenarios could incorporate multinomial random variables with more than just two possible discrete values, random variables from other discrete distributions, or continuous random variables with appropriate supports. Simulating random variables from a joint distribution with a specified covariance structure can be technically challenging for many distributions, and research on this topic is ongoing [27–29]. The method we used to apply correlated random variables to nodes in a network to achieve appropriate homophily is readily extendable to multinomial random variables. The use of continuous random variables, however, would probably require a completely different algorithmic approach and a new measure of homophily. Further work in this area would expand the types of node attributes that could be modeled, and the types of correlations and homophily that could be studied.

There are theoretical limitations to the combinations of probabilities and correlations that can be generated (S5 Fig). While the expected correlation is always 0, the range of compatible correlation values for given probabilities is not symmetric around 0 (e.g., two attributes, each with a high probability, can be strongly positively but not strongly negatively correlated). These limitations necessarily influenced the choices of probabilities and correlations we used in this study. In all analyses, we chose correlation values of equal magnitude so results could be compared in the positive and negative direction. Further, we compared correlations of magnitude 0.45 (0.15) when looking at two (three) binary attributes to ensure we investigate interaction networks where there are at least some individuals with each of the possible 4 (8) combinations of attributes. This means we studied only the effect of moderate and weak correlations. Stronger correlations will likely lead to stronger effects but exhibit the same directionality.

Homophily, the main property studied in this paper, can be considered as a type of clustering that is based on node characteristics. Small-world networks such as those used in this study also exhibit a more traditional type of "structural" clustering, measured using e.g. the local clustering coefficient [16, 30]. This type of clustering captures the idea that friends of a friend are likely to be friends and is based on edge characteristics. Most nodes in a small-world

network cannot be assigned to a single cluster as there exists high levels of overlap among the clusters. It would interesting to explore how the presence of homophily affects the spread of disease through social interaction networks with more defined structural clusters, e.g., mimicking the separation of a community into sub-communities.

Perfect isolation by those who practice social distancing led to higher $R_0$ values than very high levels of distancing (S2F Fig). Similarly, in the case of a highly effective vaccine very high vaccine coverage (80%) led to more outbreaks than slightly lower coverage (S2D Fig). Both these counter-intuitive observations only occurred in the presence of homophily, and they are likely model artifacts: The activity level of each individual corresponds to the probability that this individual is chosen as the initially infected seed case. If the activity level of distancing individuals is nonzero (e.g. 75% reduction), then there remains a small chance that an individual who distances is chosen as the seed case. If this happens, passage of the virus is unlikely in the presence of homophily, i.e. when the distancers cluster together. At perfect isolation (100% reduction), solely non-distancers, who in the presence of homophily cluster together, are chosen as seed cases. Similar reasoning explains the second observation. Note that these counter-intuitive observations did not occur when considering the relative outbreak frequency as this measure, contrary to $R_0$, takes into account the probability that an outbreak actually occurs. This is another reason why we used the relative outbreak frequency as the primary outcome measure in the generic infectious disease model.

In our model, two people who practice social distancing are less likely to interact than two people, one of whom practices social distancing and one of whom does not. This may not be a realistic assumption, as cautious individuals may avoid others who do not practice social distancing, and therefore interact more frequently with other social distancers (e.g., forming social bubbles). Detailed information regarding social distancing patterns under COVID-19 is currently lacking, however, and patterns may be heterogeneous across cultures and demographic groups. We therefore implemented a simple model for social interactions but acknowledge that our model could be extended to more realistic scenarios once more data become available.

We modeled vaccine effectiveness using an all-or-nothing approach: either the vaccine provides full protection or it has no effect. A "leaky" vaccine that reduces the infection and/or the transmission probability for all vaccinated people by a certain percentage represents an alternative approach, however the model predictions may be insensitive to how vaccine effectiveness is implemented [31]. Modeling vaccine factors such as age-varying effectiveness requires the specification of further parameters, therefore to enhance the interpretability of our model we chose not to incorporate this additional level of complexity.

## Conclusion

People interact more frequently with people of similar age and with similar attitudes or beliefs, a phenomenon known as homophily. Surveys found that individuals at high-risk for severe COVID-19 infection are more likely to practice social distancing and get vaccinated once a vaccine is available, and positive attitudes toward social distancing and vaccination seem positively correlated. The current social context in which the COVID-19 pandemic is playing out is therefore characterized by homophily and correlations among beliefs and circumstances. We developed a novel technique for generating interaction networks with several binary attributes with a defined correlation structure and defined degrees of homophily. This technique is readily extendable to multinomial random variables, and can provide a foundation for more complex studies involving attributes with more flexible probability distributions.

Using a simple generic infectious disease model as well as a more complex, previously developed model tailored specifically to COVID-19, we studied the spread of a virus in interaction networks with various levels of homophily and correlations among beliefs and circumstances. The current social context corresponds to a scenario with a substantially worse disease burden than a scenario without homophily and correlations. Differential equation models, which by default assume homogeneous mixing of the population, likely underestimate the real disease burden. Several of our study results may be relevant to policy makers, as we showed that the optimal distribution strategy of a limited vaccine depends on the relative average contact rate of low- versus high-risk individuals, as well as the level of homophily between those who are vaccinated.

## Materials and methods

### Physical interaction network

We used an agent-based disease model to study the spread of a generic infectious disease as well as COVID-19 throughout a physical interaction network of $N = 1000$ individuals, modeled as a Watts-Strogatz small-world network [16]. In these networks, each vertex represents an individual and each edge represents a possible contact between two individuals. The average degree of each vertex was $k = 14$: the average number of contacts per day per individual found in a seminal multinational study (S2 Table) [21]. We randomly rewired each edge with a probability of 5%, which is known to yield a high local clustering coefficient and low average path length, characteristic of small-world networks [16, 32].

### Modeling beliefs and circumstances

In the generic infectious disease model, we considered two binary attributes: attitude toward vaccination (1 = individual vaccinates, 0 = individual does not vaccinate), and attitude toward social distancing (1 = individual engages in risk reduction such as reduced interactions, mask wearing, increased hand washing, etc., 0 = individual does not engage in risk reduction). In the three-attribute COVID-19 model, we added risk status as a third variable (1 = individual is at high risk for COVID-19 due to age (65 and older) or known comorbidities, 0 = individual is not in the COVID-19 high-risk group).

### Generating binary attributes with a defined correlation structure

Exact forms for the distributions of correlated bivariate ($d = 2$) and trivarate ($d = 3$) Bernoulli random variables have been developed in the literature [33], and we relied on those results. We specified expectations $p_i$, $i = 1, \ldots, d$, and the covariance matrix $\Sigma$ and calculated a multivariate Bernoulli probability distribution of $d$ correlated Bernoulli random variables. In the simplest case for $d = 2$ (two binary attributes), if

$$
\begin{aligned}
p_1 &= \mathbb{E}X_1 = \mathbb{P}(X_1 = 1), \\
p_2 &= \mathbb{E}X_2 = \mathbb{P}(X_2 = 1), \\
\sigma_{12} &= \mathbb{E}[(X_1 - p_1)(X_2 - p_2)], \\
\rho_{12} &= \frac{\sigma_{12}}{\sqrt{\text{var}(X_1)\text{var}(X_2)}},
\end{aligned}
$$

where $p_i \in [0, 1]$, $\rho_{12} \in [-1, 1]$, and $\text{var}(X_i) = p_i(1 - p_i)$, the bivariate joint probability

distribution can be represented as:

$$
\begin{aligned}
P_{00} &= \mathbb{P}(X_1 = 0, X_2 = 0) = (1 - p_1)(1 - p_2) + \rho_{12}\sqrt{p_1 p_2 (1 - p_1)(1 - p_2)}, \\
P_{10} &= \mathbb{P}(X_1 = 1, X_2 = 0) = 1 - p_2 - P_{00}, \\
P_{01} &= \mathbb{P}(X_1 = 0, X_2 = 1) = 1 - p_1 - P_{00}, \\
P_{11} &= \mathbb{P}(X_1 = 1, X_2 = 1) = P_{00} + p_1 + p_2 - 1.
\end{aligned}
$$

For a third random variable, $X_3$, with random variables $X_1$ and $X_2$ as above:

$$
\begin{aligned}
p_3 &= \mathbb{E}X_3 = \mathbb{P}(X_3 = 1), \\
\sigma_{13} &= \mathbb{E}[(X_1 - p_1)(X_3 - p_3)], \\
\sigma_{23} &= \mathbb{E}[(X_2 - p_2)(X_3 - p_3)], \\
\theta_{123} &= \mathbb{E}[(X_1 - p_1)(X_2 - p_2)(X_3 - p_3)].
\end{aligned}
$$

The multivariate distribution in three dimensions is given by

$$
\begin{bmatrix}
P_{000} \\ P_{100} \\ P_{010} \\ P_{110} \\ P_{001} \\ P_{101} \\ P_{011} \\ P_{111}
\end{bmatrix}
=
\begin{bmatrix}
(1 - p_3) & -1 \\
p_3 & 1
\end{bmatrix}
\otimes
\begin{bmatrix}
(1 - p_2) & -1 \\
p_2 & 1
\end{bmatrix}
\otimes
\begin{bmatrix}
(1 - p_1) & -1 \\
p_1 & 1
\end{bmatrix}
\begin{bmatrix}
1 \\ 0 \\ 0 \\ \sigma_{12} \\ 0 \\ \sigma_{13} \\ \sigma_{23} \\ \theta_{123}
\end{bmatrix},
$$

where $\otimes$ indicates the Kronecker product of two matrices. For all simulations in this study, we considered $\theta_{123} = 0$. For $d > 3$ correlated binary attributes (not considered here), more computationally tractable algorithms have been developed [34–36], although the number of parameters required to fully specify the distribution is $2^d - 1$, and therefore grows exponentially.

One difficulty associated with generating correlated binary random variables has to do with the compatibility of the expectation vector and the covariance matrix. If $\mathbf{p} = [p_1, p_2, \dots p_d]$ is a vector of expectations for $d$ Bernoulli random variables, and $\Sigma$ is a covariance matrix, not all combinations of $\mathbf{p}$ and $\Sigma$ are compatible. For $d = 2$, an example of compatible correlation values which will result in a positive definite covariance matrix for fixed $\mathbf{p}$ is shown in S5 Fig. Explicit bounds on the correlations for the case of $d = 2$ and $d = 3$ have been derived [37]. In general, as $d$ increases, the probability that randomly chosen $\mathbf{p}$ and $\Sigma$ are compatible decreases quickly. For Figs 3, 4 and 6 (two binary attributes), where we considered a fixed proportion of individuals who get vaccinated ($p_1 = 2/3$) and who practice social distancing ($p_2 = 2/3$), the compatible range was $[-0.5, 1]$ (S5 Fig) and we considered correlation values $-0.45$, 0, and 0.45. In Fig 4 (three binary attributes), the probability an individual gets vaccinated is $p_1 = 2/3$, the probability an individual practices social distancing is $p_2 = 2/3$, and the probability an individual is high-risk is $p_3 = 1/3$. Here, the range of compatible correlation values was smaller, and we compared all combinations of second-order correlations of $-0.15$, 0, and 0.15.

## Quantifying the homophily of binary attribute assignments in an interaction network

Given an assignment of a binary attribute (opinion or circumstance) $X \sim$ Bernoulli($p$) to all $N$ vertices (individuals) in a graph (interaction network), we counted the proportion of edges (interactions) between two vertices with the same attribute value (shared opinion or circumstance). Let this proportion be denoted by $\phi \in [0, 1]$. Clearly, $\mathbb{E}[\phi] = p^2 + (1 - p)^2$. Homophily is characterized by more interactions between individuals with shared attribute values than expected by random chance. We quantified the degree of homophily using the relative clustering of the binary attribute assignment,

$$\text{homophily}(\phi) = \frac{1 - \phi}{1 - \mathbb{E}[\phi]} = \frac{1 - \phi}{1 - (p^2 + (1 - p)^2)} < 1.$$

For example, if we assume that $p = 2/3$ of all $N$ individuals get vaccinated, under a random expectation $\mathbb{E}[\phi] = 5/9$ of all interactions will involve two individuals with the same vaccination status. Therefore, $\phi = 7/9$ corresponds to 50% homophily.

The relative clustering value can be negative. However in this study we only considered non-negative levels of clustering since 'heterophily', the attraction by people with differing attribute values and/or the repulsion by people with same opinions or circumstances, seems unrealistic for the three binary attributes whose clustering effect we investigate here: who vaccinates, who practices social distancing and who is a high-risk individual. Furthermore, in a fully connected interaction network homophily is always strictly less than 1 unless all individuals have the same opinion. However, in the limit as the network size $N$ approaches infinity while the average degree remains fixed, homophily can be chosen to be arbitrarily close to 1.

## Generating opinion patterns with a defined level of homophily and a defined correlation structure

We followed a two-step procedure to obtain an assignment of $d$ binary attributes $X_i \sim$ Bernoulli($p_i$), $i = 1, \ldots, d$ to all $N$ vertices in an interaction network with a predefined correlation structure between the attributes, as well as a predefined homophily for each attribute (Fig 1B–1D).

1. To each vertex, we assigned a $d$-dimensional attribute vector with a defined correlation structure, by drawing from an appropriate multivariate Bernoulli probability distribution as described above.

2. We randomly picked one of the attributes that still exhibited lower than desired homophily. Then, we picked two vertices whose attribute vectors differed in only this attribute, and swapped their attribute values. This ensured that the correlation structure remained unchanged. We repeated this process until we reached the desired level of homophily for each attribute.

To ensure convergence of the latter process towards higher values of homophily, we defined the dissimilarity index of a vertex with respect to an attribute, denoted $d(v, a)$, as the proportion of neighbors of this vertex with a different attribute value. We then preferentially picked vertices with a high dissimilarity index to swap attribute values.

To speed up the convergence process, we used $(d(v, a))^{16}$ rather than the simple dissimilarity index $d(v, a)$ to choose which two vertices to swap attribute values. This modification prevented the algorithm from converging to homophily values lower than desired, i.e., it ensured that we quickly reached high levels of homophily such as 50%. It did however slightly modify

the resulting patterns of binary attributes as it led to fewer vertices with very high dissimilarity indices and relatively more vertices with intermediate index values. However, lower-sample-size simulations where we compared results obtained with an exponent of 16 to exponents of 1 and 4 revealed no qualitative differences in our findings (S6 Fig).

## Effect of attribute values: Vaccination, social distancing and high-risk individuals

Once attribute values were assigned to each individual, each vaccinated person was removed from the pool of susceptibles with a probability corresponding to the considered vaccine effectiveness (that is, we consider an all-or-nothing vaccine). Moreover, note that as we study the spread and potential outbreak of an infectious disease in a local community, which typically happens within a few days to weeks, we do not consider vaccination as a dynamic process. In other words, the proportion of vaccinated individuals is fixed for each simulation and vaccinations take place prior to the beginning of the simulation.

Further, each individual was assigned a base activity level $a \in [0, 1]$ describing how likely that individual was to seek contact with any neighbor in the interaction network on any given day. Throughout the paper, we used $a = 1/\sqrt{2} \approx 0.71$, assuming that all individuals have fewer physical contacts that allow disease transmission than prior to a pandemic (due to e.g. mask wearing, work-from-home orders, etc.). The activity level of those individuals who practice social distancing was further reduced by $r^{\text{distancing}} \in [0, 1]$. Note that this is a vertex-based attribute, implying that the probability of interaction between two people who practice social distancing was $r^{\text{distancing}} \cdot r^{\text{distancing}}$ lower than the probability of an interaction between two non-distancers. Also, note that while we primarily talk about social distancing throughout the paper, which can be easily understood in a network context, our abstract implementation of contacts does not require us to explicitly specify and separately model different types of risk mitigation. Rather, mask wearing, increased hand washing, social distancing, etc. all proportionately reduce the risk that a susceptible is infected through physical contact with an infectious individual, compared to a pre-pandemic level. The activity level of an individual should thus be interpreted as the combined effect of all risk mitigation efforts.

In Fig 4, we further considered the possibility of increased activity levels due to a potentially false belief of immunity following vaccination. We modeled this by introducing another vertex-based parameter $r^{\text{increase}} \in [0, 1/a - 1]$, where the base activity level of vaccinated individuals is multiplied by $1 + r^{\text{increase}}$.

In the COVID-19 model, we added risk status as a third binary attribute, and considered homophily and correlation of this attribute in addition to vaccination and social distancing. High-risk individuals have a higher chance of a symptomatic, severely symptomatic, or deadly infection [17]. In Fig 6, we considered a continuum of scenarios where all high-risk individuals practice a certain increased level of distancing, modeled by multiplying the base activity level with $r^{\text{distancing}} \in [0, 1]$ as before. In these analyses, the set of distancers coincides with the set of high-risk individuals and we did not consider differential distancing levels between individuals with the same risk status.

## Simulation of the disease spread

To simulate the spread of the generic infectious disease, we implemented a simple stochastic compartmental disease model and distinguished between susceptible (S), infected/infectious (I) and removed/recovered (R) individuals. In the COVID-19 model, the infectious compartment was split into several compartments enabling a more accurate description of the course of COVID-19 progression: exposed/pre-symptomatic (E), asymptomatic (A), symptomatic (I),

and severely symptomatic requiring hospitalization (H). To account for mortality, an additional removed compartment of individuals who have died from COVID-19 (D) was introduced [17]. In both models, time is discrete; one unit of time corresponds to one day. The simulation starts with a single seed case: a susceptible who becomes infected. In the generic infectious disease model, the seed case starts in the infected compartment, while in the COVID-19 model the seed case starts in the exposed compartment. The probability that any susceptible becomes the seed case is proportional to the activity level of that individual.

Each day, any susceptible can become infected through contact with an infectious neighbor on the interaction network. The probability of physical contact is the product of the activity levels of the susceptible and the infectious individual. If there is contact, then disease transmission occurs with transmission probability $\beta$. In the generic disease model, $\beta = 10\%$. In the COVID-19 model, $\beta$ varies over the course of the infection, is higher for symptomatic versus asymptomatic individuals and peaks at the onset of possible symptoms (details are described in [17]). Given the inherent uncertainty of COVID-19-related parameters, we sampled the transmission-related parameters from the same uniform probability distributions as in [17]: the peak transmission probability (provided contact occurs) for symptomatic individuals is $\beta_I \in \text{Unif}([5\%, 40\%])$ and for asymptomatic individuals $\beta_A \in \text{Unif}([0\%, \beta_I])$.

In the generic infectious disease model, infectious individuals eventually recover from the disease, and the per-day probability of recovery is $\gamma = 10\%$. In the COVID-19 model, infectious individuals start in the exposed compartment, while risk-group-dependent parameters and probabilities determine the transitions through the different infectious compartments. Infected individuals eventually recover or die. We used the same probability distributions and parameters governing these transitions as in the original model description (Tables 1 and 2 in [17]). In particular, we assumed that high-risk individuals have a 1 to 5 times lower rate of truly asymptomatic infections, a 4 to 10 times higher hospitalization rate (when symptomatically infected), as well as a 4 to 10 times higher death rate (when hospitalized) [38, 39]. Because of the short time frame of the simulations (weeks to months), we did not consider reinfections; recovered or dead individuals were removed from the simulation.

The more complex COVID-19 model includes additional parameters specific to SARS-CoV-2 and COVID-19, such as the proportion of asymptomatic infections and the infection fatality rate. We used the same values and sampled unknown parameters from the same uniform probability distributions as in the original model description [17].

## Outcome measures

In each simulation run of the generic infectious disease model, we recorded two outcomes: the initial basic reproductive number and a conditional outbreak probability. First, we computed the initial basic reproductive number $R_0$ as the total number of secondary infections caused by the seed case, i.e., the initially infected individual. For this calculation, if on a given day a susceptible individual was "infected" by $m \geq 1$ people where one these people was the seed case, then we added $1/m$ to the initial basic reproductive number. Second, we defined an outbreak as a situation where more than 1% of the population became infected (that is, at least 10 follow-up infectious occurred in a contact network of $N = 1000$ individuals) and recorded the proportion of cases where the infection of an initial susceptible leads to an outbreak. This can be considered a conditional outbreak probability,

$$\mathbb{P}(\text{outbreak} \mid \text{initial infection occurred}).$$

Subsequent multiplication with $\mathbb{P}(\text{initial infection occurred})$ yields a measure of the outbreak frequency.

The probability that, in a given time interval, an initial infection occurs is proportional to the total number of contacts by susceptible individuals, the community incidence rate and the virus attack rate. Since we did not make absolute predictions but instead considered relative comparisons of the outbreak frequency, only the total number of contacts by susceptible individuals matters, which is given by

$$T_{\text{susceptible}} = Nk(p_{VD}(1 - e_V)a_V a_D + p_{V\bar{D}}(1 - e_V)a_V + p_{\bar{V}D}a_D + p_{\bar{V}\bar{D}}).$$

Here, $N$ is the total number of agents in the physical interaction network, $k = 14$ is the average number of connections per individual, $e_V \in [0, 1]$ is the vaccine effectiveness (a proportion of $e_V$ vaccinated individuals is no longer susceptible), $a_D = 1 - r^{\text{distancing}} \in [0, 1]$ is the relative activity level of distancers, $a_V = 1 + r^{\text{increase}} \in [1, \sqrt{2}]$ is the relative increase in activity levels among the vaccinated population (only considered in Fig 4), and $p_{VD}, p_{V\bar{D}}, p_{\bar{V}D}$ and $p_{\bar{V}\bar{D}}$ are the proportions of the total population that are vaccinated ($V$) or not ($\bar{V}$) and practice social distancing ($D$) or not ($\bar{D}$), calculated as described in the subsection "Generating binary attributes with a defined correlation structure". The total number of contacts by all individuals is given by

$$T_{\text{all}} = Nk(p_{VD}a_V a_D + p_{V\bar{D}}a_V + p_{\bar{V}D}a_D + p_{\bar{V}\bar{D}}).$$

With this, we have

$$\mathbb{P}(\text{initial infection occurred}) = T_{\text{susceptible}}/T_{\text{all}},$$

and the main outcome metric used in Figs 2 and 3, S1–S3 Figs, the outbreak frequency, is given by

$$\mathbb{P}(\text{outbreak} \,|\, \text{initial infection occurred})\mathbb{P}(\text{initial infection occurred}).$$

In Fig 4, we considered a modified version of this outbreak frequency. Here, we analyzed the tradeoff between having a vaccine (with a certain effectiveness) and a subsequent increase in social contexts by the vaccinated. We compared the outbreak probability in the presence of a vaccine for various scenarios (specified by $e_V \in [0, 1]$ and $a_V \in [1, \sqrt{2}]$) to a situation without a vaccine, which corresponds to $e_V = 0$, $a_V = 1$, and defined this ratio as the relative outbreak frequency.

In the COVID-19 model, we recorded the total number of individuals who eventually died from the disease as an additional outcome measure. The outbreak probability and the initial basic reproductive number are unaffected by who is a high-risk individual. The severity of COVID-19 depends on risk status [20], and total mortality is the only outcome measure of the three we recorded that allows us to analyze the effect of homophily and correlation regarding risk status. Therefore, we focused solely on this measure for the COVID-19 model.

## Quantitative analysis

All model analyses were run entirely in Python 3.7. The contour plots in Figs 2 and 4, S1 and S2 Figs were generated by binning the data using a 20x20 equidistant grid, and subsequent smoothing using a 2-dimensional Savitzky-Golay filter [18]. To avoid over-smoothing, we chose a small window size of 3 and used only linear functions. Similarly, we used a one-dimensional Savitzky-Golay filter with window size 33,333 and linear functions to obtain a generalized moving average of the 333,333 data points generated for each correlation value in Fig 6 and S4 Fig.

Each data point in Figs 3 and 5, S3 and S6 Figs represents an average value across all conducted simulation runs (10,000,000 in Fig 3 and S3 Fig, and 1,000,000 in Fig 5 and S6 Fig), where those parameters shown on respective axes received a fixed value.

## Supporting information

**S1 Fig. Outbreak frequency in networks with and without homophily.** Contour plots were generated from 10,000,000 independent simulation runs with four vaccine and social distancing parameters chosen uniformly at random (axes show parameter ranges). The outbreak frequency Black (where an outbreak was defined as >1% of the population eventually becoming infected) from a reference scenario of no vaccine and no social distancing was calculated for two scenarios: Black (A-C) social interaction networks with 50% homophily of those who vaccinate and of those who practice distancing and (D-F) networks without homophily. Data was binned and smoothed using a two-dimensional Savitzky-Golay filter [18] (details in Methods). Each subplot shows the effect of variation of two parameters on the relative outbreak frequency. (A,D) vaccine coverage (x-axis) and vaccine effectiveness (y-axis), (B,E) vaccine coverage (x-axis) and proportion of those who distance, (C,F) contact reduction (in %) by those who practice social distancing (x-axis) and proportion of those who distance (y-axis). A comparison of the outbreak frequency under the two scenarios is shown in Fig 2, an equivalent analysis for the basic reproductive number in S2 Fig.
(TIF)

**S2 Fig. Basic reproductive number in networks with and without homophily.** Contour plots were generated from 10,000,000 independent simulation runs with four vaccine and social distancing parameters chosen uniformly at random (axes show parameter ranges). The basic reproductive number is shown for (A-C) social interaction networks with 50% homophily of those who vaccinate and of those who practice distancing and (D-F) networks without homophily. Data was binned and smoothed using a two-dimensional Savitzky-Golay filter [18] (details in Methods). (G-I) Comparison of the basic reproductive number in networks with and without homophily. Each subplot shows the effect of variation of two parameters on the basic reproductive number (A-F) or difference thereof between the two scenarios(G-I): (A,D, G) vaccine coverage (x-axis) and vaccine effectiveness (y-axis), (B,E,H) vaccine coverage (x-axis) and proportion of those who distance, (C,F,I) contact reduction (in %) by those who practice social distancing (x-axis) and proportion of those who distance (y-axis).
(TIF)

**S3 Fig. Effect of homophily and correlation of opinions on the basic reproductive number.** (A) The basic reproductive number ($R_0$) is compared for different scenarios regarding homophily and correlation of those who vaccinate and those who distance, and for different levels of vaccine effectiveness. (B) For each level of vaccine effectiveness, the change in $R_0$ is compared to the homogeneous case of no homophily and no correlation, which is set to 100%, respectively. (C-D) Absolute difference in $R_0$ (from A) when comparing physical interaction networks where (C) vaccinated, (D) distancers cluster (homophily = 50%) versus networks without homophily.
(TIF)

**S4 Fig. Degree to which the level of contact reduction by high-risk individuals influences vaccination priorities under various scenarios.** The average absolute mortality (first and third row) at a given additional contact reduction by high-risk individuals is shown for three different scenarios: negative (−0.45; yellow), zero (black) and positive (0.45; green) correlation between vaccinated and high-risk individuals. In addition, the relative mortality compared to

the case of no correlation (black line) is shown (second and last row). Black dashed lines and a gray triangle highlight the three intersection points of the three curves. Different situations are considered: 50% (first two rows) vs 80% (last two rows) vaccine effectiveness, 0% (first two columns) vs 50% (last two columns) homophily of those who vaccinate, and 0% (first and third column) vs 50% (second and last column) homophily of high-risk individuals. For all eight scenarios, a direct comparison of the location of the gray region in between the intersection points is shown in Fig 6C.
(TIF)

**S5 Fig. Compatible choices for the expectation and correlation of two Bernoulli random variables.** The possible range of correlations between two Bernoulli random variables with expectations $p_1$ (colors) and $p_2$ (x-axis) is shown for four fixed choices of $p_1$.
(TIF)

**S6 Fig. Robustness of the results for various exponents in the homophily algorithm.** The change in (A-C) relative outbreak frequency and (D-F) basic reproductive number $R_0$ compared to the homogeneous case of no homophily and no correlation is shown for different scenarios regarding clustering and correlation of those who vaccinate and those who distance, as well as for different levels of vaccine effectiveness. The exponent used in the homophily algorithm (see Methods) is 1 in A and D, 4 in B and E, and 16 in C and F.
(TIF)

**S1 Table. Required effectiveness for a vaccine not to yield more outbreaks given an increased activity level by those who receive the vaccine.** For different proportions of those who vaccinate and those who distance (50%, 65%, 80%) and two scenarios regarding homophily and correlation (none versus high homophily and correlation), $N = 200,000$ simulations were conducted for each considered level of increased activity by those who vaccinated (10%, 20%, 30%, 40%) with randomly chosen vaccine effectiveness, $U([0\%, 100\%])$, in addition to 200,000 simulations each without a vaccine. Using a one-dimensional Savitzky-Golay filter with window size 20,000 and linear functions, we obtained smoothed plots of the outbreak probability against the vaccine effectiveness for each increased activity level by vaccinated, and inferred the respective vaccine effectiveness (green cell values) at which the outbreak frequency under scenarios with a vaccine and increased activity levels by the vaccinated equaled the outbreak frequency without a vaccine (see the black line in Fig 4 for an example).
(TIF)

**S2 Table. Average daily contacts per country and age group.** Data from [21]. The most recently available census estimate from the United Nations Demographic Statistic Database was used for a weighted average of the contact rate across different age groups. The average contact reduction (last column) is calculated as one minus the ratio of average daily contacts by older people (fourth column) over the average daily contacts by younger people (third column).
(TIF)

## Author Contributions

**Conceptualization:** Claus Kadelka.

**Formal analysis:** Claus Kadelka.

**Methodology:** Claus Kadelka, Audrey McCombs.

**Software:** Claus Kadelka.

**Visualization:** Claus Kadelka.

**Writing – original draft:** Claus Kadelka, Audrey McCombs.

**Writing – review & editing:** Claus Kadelka, Audrey McCombs.

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
