## [Decision Letter · Decision Letter 0]

20 Oct 2021

PONE-D-21-23527Effect of homophily and correlation of beliefs on COVID-19 and general infectious disease outbreaksPLOS ONE

Dear Dr. Kadelka,

Thank you for submitting your manuscript to PLOS ONE. After careful consideration, we feel that it has merit but does not fully meet PLOS ONE’s publication criteria as it currently stands. Therefore, we invite you to submit a revised version of the manuscript that addresses the points raised during the review process. In addition to a few minor points and suggestions, the referee points to an important issue regarding the methodology and terminology used in the manuscript that the authors must carefully address. Please, provide a point by point answer and a revised manuscript by following the instructions below. 

We look forward to receiving your revised manuscript.

Kind regards,

Irene Sendiña-Nadal

Academic Editor

PLOS ONE

Journal Requirements:

Reviewers' comments:

Reviewer's Responses to Questions

**Comments to the Author**

1. Is the manuscript technically sound, and do the data support the conclusions?

Reviewer #1: Yes

2. Has the statistical analysis been performed appropriately and rigorously? 

Reviewer #1: Yes

3. Have the authors made all data underlying the findings in their manuscript fully available?

Reviewer #1: Yes

4. Is the manuscript presented in an intelligible fashion and written in standard English?

Reviewer #1: Yes

5. Review Comments to the Author

Reviewer #1: I was happy to review this paper again, being Reviewer #1 for the previous submission. In my opinion, the paper has improved in many respects, especially regarding the clarification of some terms and definitions (especially in the title), the presentations of results, and the discussion of the modeling decisions. I want to thank the authors for their careful consideration of my comments. While most of them have been addressed, there is one comment that I would like to discuss again. I also added a few minor points. Finally, I understand the authors’ wish to keep a high level of quantitative detail in the figures. Nevertheless, I took the liberty of adding a few suggestions on the figures, in case they still wish to improve their readability.

Major comment:

1. Following my previous major comment #3, I still believe the use of the term “outbreak probability” in the paper is misleading, because the authors only consider simulations starting with susceptible seed nodes; The probability they compute is in fact a conditional probability, conditional on the seed being susceptible, which is an important point to mention and explain. In any case, I still believe it would be better to consider the probability of having an outbreak when a random individual gets in contact with the virus, without conditioning on the seed being susceptible, mainly for two reasons.

First, I believe the second probability makes more sense for the problem at hand and the readers the authors want to address. Let us imagine the network being a community on an island, on which the virus arrives (maybe through a tourist visiting). I believe the authorities on the island will rather ask how likely they will have to manage an outbreak given that the virus arrived, rather than how likely they will have to manage an outbreak given that the virus arrived and that it reached a non-protected individual. Indeed, the scenarios in which the virus reaches a protected individual and nothing happens will matter to them, if there are many of them, they might deem allowing tourists safe enough.

Second, it is meaningless to compare conditional probabilities when conditions are different. Conditioning simulations on seeds being susceptible means that the number of outbreaks between simulations with different levels of vaccine effectiveness cannot really be compared. This is of course because simulations with more effective vaccines are more likely to start with anti-vaxxers, and if social distancing is correlated to vaccinating, these individuals are more likely to be sources of outbreaks.

Choosing to condition on the seed being susceptible or not probably does not affect most results in the paper, as the values examined are mostly changes in outbreak probabilities, or relative probabilities, with the number of vaccinated people and the vaccine effectiveness being constant. Considering one or the other probability is just a matter of a multiplicative factor. However, it does seem to matter in Figure 4: There, the outbreak probability (the conditional one proposed by the authors) is shown in absolute values and compared across different values of vaccine effectiveness, which seems problematic to me.

I would suggest that the authors clarify their definition of outbreak probability and revisit their results on increased activity levels on outbreak probability (Figure 4), as well as other comparisons of conditional probabilities shown in the supplementary materials (Figures S1 A and D, S2 A and D, and S3 A and B).

Minor comments:

1. I was a bit surprised to notice that the term of “agent-based model” was never used - using the term might help some readers identifying more rapidly the nature of the models.

2. I fear my previous comment on discussing the influence of picking a node proportionally to its activity level (minor comment #12 previously) was misinterpreted. I did not mean to suggest that the model specification should be altered but that the consequence of this decision could be explained to the reader, if possible (without this assumption, would current results be stronger, weaker, the same…?).

3. In the description of Figure 1, the phrase “removal of those successfully vaccinated” might still be misleading for those unfamiliar with epidemiological models, maybe simply add “from the pool of susceptible individuals”?

4. In the description of Figure 2, the subparts A, B, and C are not mentioned.

5. In the description of Figure 2, there is a typo: “¿1%”.

6. The phrase “100% = no homophily & no correlation” in Figure 3a is unclear.

7. There are a few places in the text where the term “Figure” appears without a number.

Suggestions/notes about the figures:

1. The use of red for decreased mortality and blue for increased mortality in Figures 3 and 5 might be unusual, some readers might interpret these colors the other way around (red often being negatively connotated).

2. The understanding of the figures might be improved by using colors consistently across different plots regarding the values they display.

3. Figure 4 could include a third plot where the black line shown in the first two plots (to indicate the switch between the detrimental and beneficial presence of a vaccine) is shown for different homophily/correlation situations, in order to compare them more easily.

4. In Figure 3a and 5b and 5c, the label “vaccine effectiveness” might be mistaken for the label for the gradient scale.

6. PLOS authors have the option to publish the peer review history of their article (what does this mean?). If published, this will include your full peer review and any attached files.

Reviewer #1: No

---

## [Author Response · Author response to Decision Letter 0]

16 Nov 2021

See attached pdf document response_to_reviewer_final.pdf

---

## [Decision Letter · Decision Letter 1]

22 Nov 2021

Effect of homophily and correlation of beliefs on COVID-19 and general infectious disease outbreaks

PONE-D-21-23527R1

Dear Dr. Kadelka,

We’re pleased to inform you that your manuscript has been judged scientifically suitable for publication and will be formally accepted for publication once it meets all outstanding technical requirements. 

When you receive the proofs please consider the suggestion made by the Reviewer about the use of the term "agent-based network". 

Kind regards,

Irene Sendiña-Nadal

Academic Editor

PLOS ONE

Additional Editor Comments (optional):

Reviewers' comments:

Reviewer's Responses to Questions

**Comments to the Author**

1. If the authors have adequately addressed your comments raised in a previous round of review and you feel that this manuscript is now acceptable for publication, you may indicate that here to bypass the “Comments to the Author” section, enter your conflict of interest statement in the “Confidential to Editor” section, and submit your "Accept" recommendation.

Reviewer #1: All comments have been addressed

2. Is the manuscript technically sound, and do the data support the conclusions?

Reviewer #1: Yes

3. Has the statistical analysis been performed appropriately and rigorously? 

Reviewer #1: Yes

4. Have the authors made all data underlying the findings in their manuscript fully available?

Reviewer #1: Yes

5. Is the manuscript presented in an intelligible fashion and written in standard English?

Reviewer #1: Yes

6. Review Comments to the Author

Reviewer #1: I thank the authors for their careful consideration of my comments and the extremely thorough corrections they made to their manuscript. I believe it shows commendable scientific dedication. I therefore recommend this article to be published and am looking forward to seeing it in this journal.

There is only one tiny detail I wanted to mention, and it is the use of the term "agent-based" made in the corrected manuscript for the "physical interaction network" (see methods section). I believe agent-based modellers would usually not refer to a network as being "agent-based", but rather to the disease model on this network as being an agent-based model (in oppoosition to differential equation models). If the authors agree with me, it would be a very quick adjustment.

I wish the authors best of luck in their current and upcoming research.

7. PLOS authors have the option to publish the peer review history of their article (what does this mean?). If published, this will include your full peer review and any attached files.

Reviewer #1: No

---

## [Editor Report · Acceptance letter]

25 Nov 2021

PONE-D-21-23527R1 

Effect of homophily and correlation of beliefs on COVID-19 and general infectious disease outbreaks 

Dear Dr. Kadelka:

I'm pleased to inform you that your manuscript has been deemed suitable for publication in PLOS ONE. Congratulations! Your manuscript is now with our production department. 

Kind regards, 

on behalf of

Dr. Irene Sendiña-Nadal 

Academic Editor

PLOS ONE